# Prevention of the Retrogradation of Glutinous Rice Gel and Sweetened Glutinous Rice Cake Utilizing Pulsed Electric Field during Refrigerated Storage

**DOI:** 10.3390/foods11091306

**Published:** 2022-04-29

**Authors:** Shuang Qiu, Alireza Abbaspourrad, Olga I. Padilla-Zakour

**Affiliations:** Department of Food Science, Cornell University, Stocking Hall, Ithaca, NY 14853, USA; sq59@cornell.edu (S.Q.); alireza@cornell.edu (A.A.)

**Keywords:** glutinous rice grain, starch retrogradation, pulsed electric field (PEF), texture profile analysis (TPA), X-ray diffraction, differential scanning calorimetry (DSC)

## Abstract

Pulsed electric field (PEF) processing is an emerging non-thermal technology that shows the potential to improve food quality and maintain stability. However, the attributes and retrogradation properties of food products made of PEF-treated rice grains are still unknown. In the current study, glutinous rice gels (GR-G) and sweetened glutinous rice cakes (GR-C) made of PEF-treated rice grains were prepared and investigated during 14 days of storage at 4 °C. The hardness values of both the GR-G and GR-C-control samples, respectively, increased from 690 g to 1423 g and from 720 g to 1096 g; the adhesiveness values of the GR-G-control and GR-C-control samples decreased to the range of −7.2 g s to −10.0 g s during storage. PEF-treated samples (3 kV/cm, 400 pulses) resulted in preventing effects against retrogradation, resembling the original textural values of the freshly prepared control samples. The high intensity of imposed PEF treatment (300–400 pulses) significantly reduced the gelatinization enthalpy values of both GR samples to 0.3–0.7 J/g. The diffraction patterns of PEF-treated GR samples were analogous to the amorphous peak of fresh-made rice gel. FTIR results indicated that PEF-treated rice grains presented fewer crystalline regions and a lesser extent of the organized double helices after refrigerated storage.

## 1. Introduction

Glutinous rice (*Oryza sativa*), also called waxy or sticky rice, is cultivated in most Asian countries, and glutinous rice grain or glutinous rice flour is commonly utilized as the main ingredient in many desserts, snacks and staple foods. For example, glutinous rice cake (Mochi) is made from glutinous rice flour (or rice grains) and other food ingredients, including sugar, vegetable oil, food thickener or stabilizer, etc. [1]. As glutinous rice starch mainly consists of amylopectin (>98%), glutinous rice grain or flour provides a product with an adhesive texture and longer shelf life against starch retrogradation than normal (amylose-contained) rice grains [2]. Retrogradation of starchy products is one of the major causes of significant food waste in the global food industry. However, glutinous rice products could still become crystallized and staled during long-term (>14 days) refrigerated storage. Our previous study confirmed that moisture content was the most significant factor to prevent glutinous rice grain retrogradation [3]. Negligible retrogradation attributes were found in cooked rice grains during a storage time of 14 days at 4 °C if the moisture content was above 80% [3]. Moreover, the addition of sugar, NaCl, lipid and surfactant significantly restrained the retrogradation extent of rice grains after 14-day storage at 4 °C [3]. Nonetheless, the longer shelf life (>14 days) of glutinous rice products for up to one to three months is worthy of investigation because of the market demands.

As a non-thermal emerging technology, pulsed electric field (PEF) treatment can be utilized to deactivate microorganisms and fruit/vegetable enzymes [4,5], improve the efficiency of food dehydration [6] and promote the juice yield and extraction of functional compounds from fruits and vegetables [7]. Compared to conventional thermal processing, PEF is able to protect food quality, such as less degradation of nutritional components and better preservation of organoleptic attributes [8].

The effects of PEF treatment on physicochemical properties of raw glutinous rice grain and its starch were reported in our previous study [9]. PEF treatment was able to change the microstructure of rice components and induce physicochemical properties changes in starch molecules, which provided the potential for extending the shelf life of glutinous rice products during refrigerated storage [9]. However, the gel textural properties, retrogradation behaviors and stability of cooked glutinous rice grains with PEF treatment have not been investigated yet. In addition, the functionality of food products made of PEF-treated rice grains is still unknown. The retrogradation properties of commercial food products made of PEF-treated rice grains have not been reported.

The objectives of the current work were to evaluate the effects of PEF treatment on the stability and retrogradation behaviors of glutinous rice samples during long-term storage at 4 °C. It has been confirmed that a longer treating time (300–400 pulses) and more pulses at a moderate PEF field strength (less than 5 kV/cm) could modify the physicochemical properties of rice grains [9]. In this study, steamed glutinous rice grains were processed for 100 to 400 PEF, treating pulses at the field strength of 3 kV/cm. The glutinous rice gels prepared with pre-cooked rice grains with or without PEF treatment were investigated. Moreover, sweetened glutinous rice cake samples, made of rice grains treated with or without PEF treatment, sugar, vegetable oil and food-grade gum, were also evaluated. The textural properties and retrogradation behaviors of glutinous rice samples were studied during long-term refrigerated storage. Results from the current study will add more knowledge to the processing of glutinous rice products, and this study will also facilitate the utilization of PEF technology in industrial food processing.

## 2. Materials and Methods

### 2.1. Materials

Commercial polished glutinous rice grains (from Thailand), food-grade sucrose and vegetable oil were obtained from a local supermarket in Ithaca, NY, USA. Kappa-carrageenan (Ticaloid^®^ 710H Powder) was kindly supplied by TIC Gums Inc. (Belcamp, MD, USA) as a gift. Potassium sorbate (P1592-45) was purchased from ChemProducts Inc. (Tualatin, OR, USA).

### 2.2. Preparation of Glutinous Rice Gel (GR-G) Samples and Sweetened Glutinous Rice Cake (GR-C) Samples Using Glutinous Rice Grains (GR) with or without PEF Treatment

Distilled water soaking, steam pre-cooking and water-bath cooking methods were applied to GR to prepare adequately gelatinized GR-G and GR-C samples (Figure 1). Soaking and steam heating were able to pre-gelatinize GR and facilitated the application of PEF on whole rice grains according to our previous study [9]. The mixed solutions of kappa-carrageenan, sugar and oil were prepared in advance. Kappa-carrageenan solution (0.7%, *w*/*w*, wet basis) was prepared by slowly adding carrageenan powder to distilled water with vigorous stirring using a magnetic stirrer, then the capped flask of gum slurry was heated to 75 °C and gently stirred at 45 °C for 12 h using a Cimarec stirring hotplate (Barnstead Thermolyne Corp., Ramsey, MN, USA) to produce a homogeneously hydrated solution. Carrageenan solution was applied to avoid the sediment of rice grains, which could contribute to having a homogenous and consistent texture for the final gel products. The mixed ingredient solutions of GR-C samples were prepared by adding sucrose and vegetable oil to a well-dissolved carrageenan solution and then the mixture was continuously stirred at 45 °C for 2 h. The concentrations of carrageenan (0.7%, *w*/*w*, wet basis), sucrose (11%, *w*/*w*, wet basis) and vegetable oil (2%, *w*/*w*, wet basis) were optimized and applied to obtain a moderate sweet taste and a characteristic glutinous rice gel texture of the final GR-C products. Potassium sorbate (0.1%, *w*/*w*) was added to each sample solution as a preservative to avoid spoilage in the final product during refrigerated storage.

Dry raw GR were mixed and soaked in adequate amounts of distilled water at room temperature for 2 h at the ratio of 1:2 (rice to water, *w*/*w*) to pre-hydrate the raw rice grains [3]. Afterward, soaked grains were drained and washed twice using distilled water. Soaked rice was pre-cooked using a steaming oven (Cuisinart^®^ Steam and Convection Oven, Cuisinart, Stamford, CT, USA) at 98.9 °C for 35 min. Steamed GR was immediately cooled in the ice water bath for 3 min, and the cooling procedure also dispersed and separated the rice grains. A lab-scale PEF PilotTM unit (Elea GmbH, Quakenbrück, Germany) was used to treat pre-cooked GR. Steamed GR (200 g) was resuspended in 200 g of 6 mM KCl solution to ensure the suspension had an adequate conductivity. The same weight of ice (200 g) was substantially added to the rice suspension to retain the temperature of the mixture below 50 °C during PEF treatment. A rectangular chamber {20 cm (L) × 10 cm (W) × 20 cm (H)} was used to hold the rice suspension during PEF treatment.

The field strength of the PEF treatment applied was 3 kV/cm. Pulse duration and frequency were 40 μs and 1 kHz, respectively. PEF treatments were conducted on GR suspensions from 100 to 400 pulses. Input energy was calculated based on Equation (1) provided by Elea GmbH (Quakenbrück, Germany). Hence, 100, 200, 300 and 400 pulses of PEF treatment could result in a total input energy of 112.5, 225, 337.5 and 450 kJ/kg, respectively.
Energy intake per pulse = 0.5∙***C***∙*U*^2^(1)
where *U* is voltage (V); ***C*** is the capacity of the PEF unit. As for the PEF PilotTM unit, ***C*** = 1.0 μF.

The initial and final temperatures and conductivity of the sample suspensions were measured using a 35100-K AquaTuff Type-K thermocouple (Cooper-Atkins Corp., Middlefield, CT, USA) and a Type 700 conductivity meter (Chemtrix BV, Limburg, The Netherlands), respectively. A portion of rice grains was frozen using liquid nitrogen immediately after PEF treatment and was freeze-dried using a freeze drier (In-Home HarvestRight, Harvest Right Inc., Salt Lake City, UT, USA) for microstructure observation.

The same amount of PEF-treated rice grains (232.5 g) was either mixed with carrageenan solution (177.5 g) or with blended carrageenan solution containing sucrose and oil (242.5 g) to prepare the GR-G or GR-C samples. In the final GR samples, GR-G samples were prepared in the same ratio of rice grains/water as that in the GR-C samples to compare the retrogradation behaviors of two batches of GR samples. Mixed PEF-treated GR and carrageenan slurries were stuffed in cylindrical plastic containers (2.2 cm in diameter, 25 cm in height) to ensure a uniform shape and appearance of the final products. Sealed containers were immersed in a water bath at 95 °C and heated for 60 min to adequately gelatinize rice grains. Cooked samples in sealed containers were then equilibrated and cooled at room temperature for 2 h. Refrigerated storage is the typical way to evaluate the accelerated retrogradation behaviors and textural changes of waxy rice starch and glutinous rice [10,11]. Therefore, rice samples were then stored at 4 °C for 14 days for further measurements.

### 2.3. Water Content of GR-G and GR-C Samples

The moisture content of the GR-G and GR-C samples with or without PEF treatment was determined using the oven-drying method according to [3].

### 2.4. Scanning Electron Microscopy (SEM) Observation of GR Surface with or without PEF Treatment

Freeze-dried GR samples were equilibrated overnight in a desiccator. Samples were fixed on aluminum studs using a double-sided adhesive tape and then were sputter-coated with gold using a TED PELLA Cressington Sputter Coater (TED PELLA, Inc., Redding, CA, USA). The surface microstructure of rice grain samples with or without PEF treatment was observed using a JCM-6000 microscope (JEOL USA, Inc., Pleasanton, CA, USA). The sputter-coated samples were transferred to a microscope mount at an acceleration voltage of 15 kV and observed at a magnification of 500×.

### 2.5. Textural Analysis

Textural properties of the GR-G samples and GR-G samples were determined using a TA-XT2 texture analyzer (Stable Micro Systems, Godalming, UK) according to Huang et al. [12]. Both GR-G and GR-C samples were molded to a cylindrical shape, with a height of 2.0 cm and a cross-sectional area of approximately 3.5 cm^2^ to acquire consistent textural profiles. Five sample replicates were tested and their average values with standard deviation (SD) were calculated.

### 2.6. Differential Scanning Calorimetry

After 4 °C storage for 14 days, a portion of the GR-G and GR-C samples with or without PEF treatment were freeze-dried. Freeze-dried rice grains were ground using a high-speed mixer at 15,000 rpm for 3 min, and the milled rice flour was passed through a 125-μm sieve. The retrogradation properties of milled rice samples were measured using a Q2000 Modulated DSC (TA Instruments, New Castle, DE, USA) according to Khanna and Tester [13], with minor modifications. Well-mixed 2.5 ± 0.1 mg of rice powder samples and 7.5 µL of water were hermetically sealed in an aluminum pan. The prepared pans were measured from 20 to 130 °C at a heating rate of 10 °C/min under a continuous flow (60 mL/min) of dry N2 gas and then cooled to 20 °C at the same rate. The retrogradation enthalpy (Δ*H_ret_*) was analyzed using TA Universal Analysis Q2000 software (version: v3.2.07). Each test was performed three times and their average values with SD were calculated.

### 2.7. Fourier Transform Infrared Spectroscopy

Thirty (30) mg of freeze-dried rice flour were analyzed using IRAffinity-1S Fourier Transform infrared spectroscopy (SHIMADZU Corp., Kyoto, Japan) equipped with a Quest attenuated total reflectance (ATR) accessory (Specac Company, Orpington, UK) from 400 to 4000 cm^−1^. Measurements were performed at room temperature and 47% relative humidity. Signal averages were obtained from 32 scans at a resolution of 4 cm^−1^. An interactive baseline correction procedure was applied.

### 2.8. X-ray Diffraction (XRD)

The crystalline study of freeze-dried rice flour samples after 4 °C storage for 14 days with or without PEF treatment was carried out using a Rigaku SmartLab X-ray diffractometer (Rigaku Denki Co., Ltd., Tokyo, Japan) operating at 40 kV and 25 mA, producing Cu/Kα radiation of a 0.154 mm wavelength (Cairns, et al., 1990). The freeze-dried samples were stored in desiccators overnight before testing. Diffractograms were obtained by scanning at the scattering angle from 4° (2θ) to 40° (2θ) at an increment of 0.5°/min, a step size of 0.02°, a divergence slit width of 1°, a receiving slit width of 0.02 mm, and a scatter slit width of 1°. Each sample was measured twice.

### 2.9. Data Analysis

The reported results were obtained using the SAS system (Version 9.1 for Windows, SAS Institute Inc., Cary, NC, USA). Analysis of variance (ANOVA) was used to determine significant differences between the results. Duncan’s test was used to separate the mean with a significance level of 0.05.

## 3. Results and Discussion

### 3.1. Proximate Analysis and Microstructure Properties of Glutinous Rice Grains with or without PEF Treatment

The temperature and electric conductivity of rice grain suspensions before and after PEF treatment are reported in Table 1. According to our previous study [9], an ice water/KCl solution mixture was applied to balance the PEF-induced temperature increase and control the final temperature. KCl solution (6 mM) was used to provide a conductive dispersion that ensured the transmission of electric pulses toward rice grains [14,15]. In Table 1, more pulses of PEF treatment led to a higher suspension final temperature, and it also contributed to larger conductivity values due to the promotion of ions movement after PEF treatment. For that case, the temperature was 55.2 °C after PEF treatment at 3 kV/cm for 400 pulses. This particular temperature could not alter the thermal behaviors of GR as they were pre-gelatinized during steaming. The raised temperature in uncooked GR suspension after PEF treatment was also reported in our previous studies, and this effect was caused by ohmic heating, which was independent of substances and sample status [9]. Due to the higher temperatures of the PEF chamber and electrodes, the initial conductivity of GR suspensions in posterior batches (PEF-300 pulses and PEF-400 pulses) was higher than the control due to part of the ice melt at the beginning.

The fresh-made GR-C-control sample (63.1%) had a lower moisture content than the GR-G-control sample (68.8%) as the portions of sugar and oil ingredients (about 13.7%, *w*/*w*) replaced the water fraction during the GR-C sample preparation (Table 1). Moreover, the moisture content of the GR-G and GR-C samples made of the rice grains treated with 400 pulses at 3 kV/cm increased from 68.8% to 76.3% and 63.1% to 68.4%, respectively, compared to the control samples. These results indicated that PEF treatment improved the water absorption of pre-cooked GR, in agreement with our previous study that PEF promoted GR water absorption, the interaction of water/rice components and hydration of rice starch in raw glutinous rice grains [9].

### 3.2. Microstructure of Pre-Gelatinized Glutinous Rice Grains with or without PEF Treatment

SEM images of pre-treated rice grain surfaces with or without PEF treatment are presented in Figure 2. Rice starch particles had been gelatinized during steaming, which has been confirmed by DSC measurement as no endothermic peak was found in the DSC thermogram (data not shown). Thus, unlike the compact structure and tight arrangement of the raw polished rice surface [9], smooth starch gel fragments were observed on the surface of rice grains in all SEM images (Figure 2A–E). On the surface of the control sample (Figure 2A), a few starch particles showed polyhedral shapes, which were typical for raw waxy rice starch particle appearance, as a result of insufficient water hydration or insufficient contact with steam. Shallow pores and cavities were also found on the surface of the control rice grain due to water absorption during the soaking and steaming processes [9].

On the other hand, more well-defined and deeper pores were observed embedded on the rice surface as the imposed intensity of PEF treatment was increasing (Figure 2B–E). The PEF electroporation effects on steamed/pre-cooked rice grains observed in the present study were similar to the surface change of PEF-treated uncooked rice grains [9]. Those pores could be attributed to the structural changes of starch and electroporation-sensitive rice proteins during PEF treatment [9]. Moreover, homogeneous smooth gelling structures were observed at the surface of PEF-treated GR samples (Figure 2B–E) without showing polyhedral-shaped starch particles. This result confirmed that starch hydration and gelatinization were promoted by100–400 pulses of PEF treatment at 3 kV/cm, which contributed to more swelling and gelling of starch particles. This finding was also in agreement with the shape transformation and promoted the hydration of potato starch, corn starch and cassava starch, which were subjected to PEF treatment at a field strength of 30 and 40 kV/cm [15,16].

### 3.3. Texture Profile Analysis

The hardness, adhesiveness, springiness and cohesiveness characteristics of the GR-G and GR-C samples after 4 °C storage for 0 or 14 days are shown in Figure 3. Fresh-made GR-G samples made of rice grains with PEF treatment at an increasing intensity from 100 to 400 pulses (sample GR-G-1 to GR-G-4) had smaller hardness values (from 690.4 g to 549.0 g) compared to the control sample (GR-G-control, 694.1 g). This result indicated that PEF treatment softened the texture of GR-G samples. Similar results were also observed in GR-C samples prepared using PEF-treated rice grains (sample GR-C-1 to GR-C-4), as the hardness values of GR-C-1 to GR-C-4 samples were smaller than the control sample (GR-C-control). Changes in gel firmness induced by PEF treatment could be associated with increased water content in PEF-treated samples (Table 1), which indicated that the enhanced hydration extent of GR led to a softer gel structure. It was noteworthy to observe that the hardness values (ranging from 549.0 g to 690.4 g) of fresh-made PEF-treated GR-G samples (GR-G-1 to GR-G-4) were, respectively, larger than the hardness values (ranging from 420.5 g to 630.5 g) of fresh-made PEF treated GR-C samples (GR-C-1 to GR-C-4), while the hardness values of the fresh-made GR-G-control (690 ± 20 g) and GR-C-control (720 ± 94 g) were insignificantly different (*p* < 0.05). This finding might be due to rice starch and rice grains yielding a weaker gelling texture in the presence of sugar and oil during the water-bath cooking compared to rice grains alone [17].

As shown in Figure 3, the adhesiveness values of both the fresh-made GR-G and GR-C samples with PEF treatment were insignificantly different from the GR-G-control and GR-C-control, respectively. However, the adhesiveness values of all fresh-made GR-G samples (from −90.2 g∙s to −110.6 g∙s) were found to be significantly (*p* < 0.05) larger than that of the fresh-made GR-C samples (from −15.6 g∙s to −25.9 g∙s), which indicated that the addition of sugar and oil dramatically decreased the adhesiveness values of rice cake samples (GR-C-control, GR-C-1 to GR-C-4) compared to rice/water only samples (GR-G). It has been reported that the presence of disaccharides decreased the stickiness of gelatinized starch samples, and disaccharides/oil also reduced the abrasion between gelatinized starch granules [3]. As for the springiness and cohesiveness of freshly made rice samples, PEF-treated samples (GR-G-1 to GR-G-4, GR-C-1 to GR-C-4) showed similar results of springiness and cohesiveness compared to the control samples (GR-G-control and GR-C-control). The addition of sugar and oil decreased the values of springiness from the ratio of 0.68–0.73 (GR-G samples) to the ratio of 0.50–0.54 (GR-C samples), respectively, and the values of cohesiveness from 0.32–0.36 (GR-G samples) to 0.27–0.29 (GR-C samples), respectively. This result also confirmed that the presence of sugar and oil in rice gels depressed the growth of gel-like structures and affected the integrity of fresh rice gels.

After 4 °C storage for 14 days, the hardness values of both the GR-G and GR-C-control samples significantly increased from 690 g to 1423 g and from 720 g to 1096 g, respectively. The higher gel firmness is mainly caused by starch retrogradation, which is associated with the syneresis of the starch gel and recrystallization of amylopectin during storage [18]. The higher input energy of PEF treatment resulted in a lesser extent of hardness increment in the rice gel samples of GR-G-1 to GR-G-4 and rice cake samples of GR-C-1 to GR-C-4 after storage. PEF treatment at 3 kV/cm for 400 pulses (GR-G-4 and GR-C-4) resulted in the most pronounced preventing effect against retrogradation, resembling the original values of the freshly prepared control sample. The reason that PEF-treated samples showed better stability against refrigerated storage could be associated with the increased moisture content in the PEF-treated rice gel samples of GR-G-1 to GR-G-4 (69.2% to 76.3%), and PEF-treated rice cake samples of GR-C-1 to GR-C-4 (63.5% to 68.4%) (Table 1). Water content is one of the primary factors that influence starch retrogradation behaviors during storage. According to previous studies, the relationship between water content and starch retrogradation displayed a bell curve, with the maximum extent of retrogradation of starch gels determined at 40% to 45% water content [19,20]. Glutinous rice gels showed negligible retrogradation behaviors if the water content was above 80% during refrigerated storage for 14 days in our previous study [3]. In the current study, 69.2% to 76.3% water content of the GR-G gels and 63.5% to 68.4% water content of the GR-C samples made of PEF-treated glutinous rice grains showed satisfactory consistency, backed by the texture values, and delivered similar texture to the fresh control after 14-day storage at 4 °C (Figure 3A,B). In contrast, control samples (GR-G-control and GR-C-control) showed crystallized and grainy textures after refrigerated storage (Figure 3A,B).

Steam pretreatment, PEF treatment and water-bath heating stepwise contributed to the structural changes of starch and protein in fresh-made GR samples. Once the gelatinization in excess water is initiated, it continues until the starch granules are completely disrupted, which relates to both the disorganization of starch chemical structures and physical swelling/rapture of starch granules [11,19]. Therefore, the GR samples with PEF treatment after refrigerated storage for 14 days showed slight retrogradation behaviors at the moisture content from 63.5% to 76.4%, likely due to the promotion of water hydration, swelling and structural disruption in PEF-treated rice grains. Moreover, the hydroxyl groups of sugars were able to form hydration bonds with starch and water molecules, which prevented the formation of hydrogen bonds among starch molecules and recrystallization during retrogradation [21,22]. Lipids could form a complex with the external branch structures of amylopectin or amylopectin short chains and thereby inhibited starch long-term retrogradation [20]. In our current study, PEF could promote the interactions of sugar-starch and oil-starch in rice cakes that jointly contributed to good stability in the GR-C-1 to GR-C-4 samples at a relatively low water content range (63.5% to 68.4%) after refrigerated storage of 14 days. These results also confirmed the role of sucrose and vegetable oil in preventing the retrogradation behaviors of glutinous rice [3].

Starch retrogradation results in changes in the textural parameters, including a hardness increase or adhesiveness decline [21]. The adhesiveness values of the GR-G-control and GR-C-control samples decreased to a relatively low level (−7.2 g∙s to −10.0 g∙s) after storage for 14 days (Figure 3). According to our previous study [3], typical and measurable recrystallization and retrogradation behaviors of glutinous rice samples with 68–70% water content were determined by TPA measurements after 14-day storage at 4 °C (Figure 3); hereby the 68% water content of rice gel was designed as the GR-G-control sample in the current study. Moreover, the addition of sucrose and vegetable oil was proven to be an effective method to prevent starch retrogradation, and the GR-C-control sample showed better stability than the GR-G-control sample, but the GR-C-control lost its edible texture as the adhesiveness value was >−10 g∙s. Similar to the hardness results, the increasing intensity of PEF treatments was able to impede adhesiveness decline in both the GR-G (GR-G-1 to GR-G-4) and GR-C samples (GR-C-1 to GR-C-4).

The springiness values of the GR-G-control and GR-C-control samples decreased, and the cohesiveness values of the GR-G-control and GR-C-control samples increased after storage for 14 days. PEF treatment for 100–400 pulses at 3 kV/cm showed a significant influence in inhibiting the changes in springiness and cohesiveness values of the GR sample gels during storage. This result provides further evidence that PEF-treated rice samples showed good stability against long-term retrogradation.

### 3.4. Differential Scanning Calorimetry (DSC) Analysis

It has been reported that the PEF treatment-induced structural changes of amylopectin in native glutinous rice grains could result in lower values of thermal characteristics, including *T_o_* and Δ*H_gel_* values, during gelatinization. The crystalline structure of amylopectin in raw rice grains could be partially disrupted and gelatinized during PEF treatment, requiring less energy to deconstruct the hydrogen bonds during gelatinization [9]. In the current study, gelatinization characteristics were not present since rice grains were gelatinized during steaming.

The values of Δ*H_ret_*, calculated from the DSC measurement, represented the melting enthalpy of double helices or recrystalline structures of amylopectin formed during long-term storage [21]. Therefore, the larger value of Δ*H_ret_* indicated a further tendency of starch retrogradation after storage. As shown in Figure 4, both the GR-G and GR-C samples made of increasing PEF pulses (100 to 400) of treated rice grains presented significantly (*p* < 0.05) lower Δ*H_ret_* values and a lesser extent of retrogradation than the respective control samples after 14-day storage at 4 °C. The high intensity of imposed PEF treatment (300–400 pulses) significantly reduced the Δ*H_ret_* values of both the GR-G and GR-C samples to 0.3–0.7 J/g, and the low level of amylopectin recrystallized enthalpy confirmed the retrogradation preventing effects induced by PEF treatment. Higher moisture content in 400 pulses PEF-treated samples could promote stretching and extending in amylopectin structures in rice grains during gelatinization, which prevented the recrystallization during storage.

The GR-G-control sample showed a higher value of Δ*H_ret_* than the GR-C-control and other PEF-treated GR samples. It has been indicated that glutinous rice gel with the addition of NaCl, sucrose, lipids and surfactants (monoglycerides and sucrose ester) showed a lesser extent of recrystallization compared to the water/GR-only sample after storage [3]. This result confirmed that GR mixed with sugar and lipid showed better stability than the GR-only sample during storage, and PEF treatment enhanced the inhibiting effects of sugar and lipid against starch retrogradation.

### 3.5. X-ray Diffraction (XRD) Pattern

Powdered samples of GR-G and GR-C after 14 days of storage at 4 °C were scanned using XRD to determine the amorphous/crystalline status of the starch. It has been reported that native glutinous rice flour showed an A-type diffraction pattern during XRD measurement [9,21]. The A-type crystallinity peaks will diminish and become smooth during heating, with the fresh-gelatinized rice flour presenting an amorphous peak while losing its crystallinity as the result of starch gelatinization [23]. Therefore, the fresh-cooked GR-G-control sample showed an amorphous broad peak in the range from 2θ = 17.3° to 21.6° (Figure 5A). The diffraction peaks of the control GR-G sample (GR-G-control) were shifted to a higher scattering angle after 4 °C storage for 14 days, which confirmed the structural transformation due to starch recrystallization. Partially recrystallized peaks were defined as the ‘B-type’ diffraction pattern [23]. According to Cairns et al. [24], the XRD peak area increased during storage as retrogradation occurred in starch gels. Compared to the GR-G-control sample, much smaller and narrower peaks of samples made of 200–400 pulses PEF-treated rice grains (GR-G-2 to GR-G-4) were observed in Figure 5A. In addition, the XRD patterns of the GR-G-2 to GR-G-4 samples were similar to fresh-made GR-G samples (GR-G-control-Freshly Made) without showing an apparent peak shift. The shape difference and peak position in XRD patterns, which represented different retrogradation degrees of the starch gel, indicated that PEF-treated GR-G samples showed less retrogradation extent than the control.

The GR-C-control sample showed a lesser extent of peak shift and smoother peaks than the GR-G-control sample (Figure 5B), which correlated with the results that the addition of disaccharide and lipid could prevent the retrogradation of rice gel in Section 3.2. Similar amorphous peaks were found in the diffraction patterns of GR-C-1 to GR-C-4 samples, which were analogous to the amorphous peak of the fresh-made rice gel (GR-G-control-Freshly Made). In fact, both the GR-G and GR-C samples made of 400 pulses of PEF-treated rice grains (GR-G-4, GR-C-4) showed little extent of crystallization after refrigerated storage of 45 days (data not shown), which was a promising result for the industrialization of shelf-stable glutinous rice products. The XRD results of rice samples after storage were in agreement with the retrogradation enthalpy data reported in Section 3.3.

### 3.6. FTIR Spectrum of GR Samples

Figure 6 depicts the FTIR spectra of GR samples after refrigerated storage for 14 days. According to our previous study, PEF treatment promoted the partial gelatinization of rice starch without introducing any new chemical groups and molecular dissociation [9]. Structural and conformational changes, due to starch retrogradation, can be monitored by the analyses of FTIR band shapes and peak intensity of conformational-sensitive bands in the wavelength region of 1300–800 cm^−1^ [21]. In our current study, it was observed that the band intensity of the control samples in the region of 1300–800 cm^−1^ was weaker than in PEF-treated GR samples after refrigerated storage (Figure 6A,B). In the disordered state of fresh-gelatinized starch paste, starch molecules have spread conformations. As the starch system becomes more ordered during storage, the level of spread conformations will be reduced, resulting in a smaller distribution of chemical bond energies compared to the initial gelatinized state [25]. Hereby, the band-narrowing phenomenon was observed in retrograded starch molecules [9]. Therefore, more pronounced bands of PEF-treated GR samples (GR-G-1 to GR-G-4, and GR-C-1 to GR-C-4) were observed in contrast to the control samples (GR-G-control and GR-C-control). The bands of all GR-C samples (GR-C-control, GR-C-1 to GR-C-4) showed more pronounced patterns than all GR-G samples (GR-G-control, GR-G-1 to GR-G-4), which confirmed the result that adding sugar and lipid was effective in preventing the rice retrogradation compared to a higher amount of moisture content in GR-G samples.

It has been illustrated that the peaks at 1047 cm^−1^ and 1022 cm^−1^ reflected the characteristics of the starch crystalline region and amorphous starch, respectively [26]. During refrigerated storage, starch retrogradation has been observed to cause an increment in the ratio of peak intensities at 1047 and 1022 cm^−1^, which suggests a reduction in amorphousness or an increase in ordered structure [25]. In this study, PEF treatment diminished the peak at 1047 cm^−1^ and enhanced the band at 1022 cm^−1^ (Figure 6). The lower absorbance ratios of 1047/1022 cm^−1^ and 1022/995 cm^−1^ showed that PEF-treated rice grains presented fewer crystalline regions and a lesser extent of the organized double helices [21].

## 4. Conclusions

We investigated the application of PEF (100, 200, 300 and 400 pulses) at moderate field strength (3 kV/cm) to pre-cooked whole glutinous rice grains, to prevent or mitigate retrogradation of rice gels (made with glutinous rice grains and gum) and rice cakes (made with glutinous rice grains, sugar, oil and gum) during 14-day refrigerated storage. PEF treatments created pores at the rice surface and PEF also allowed rice samples to retain more moisture, and better diffusion of sugar, oil and gum, resulting in products with significantly less retrogradation during refrigerated storage, as confirmed by microscopy, TPA, XRD, DSC and FTIR analyses. PEF, under the conditions tested, showed very good potential to develop refrigerated and shelf-stable glutinous rice products with better texture stability. Further studies focused on sensory evaluation and consumer acceptability of PEF-treated glutinous rice products, as well as associated costs, will determine the technical and commercial feasibility of this technology.

## Figures and Tables

**Figure 1 foods-11-01306-f001:**
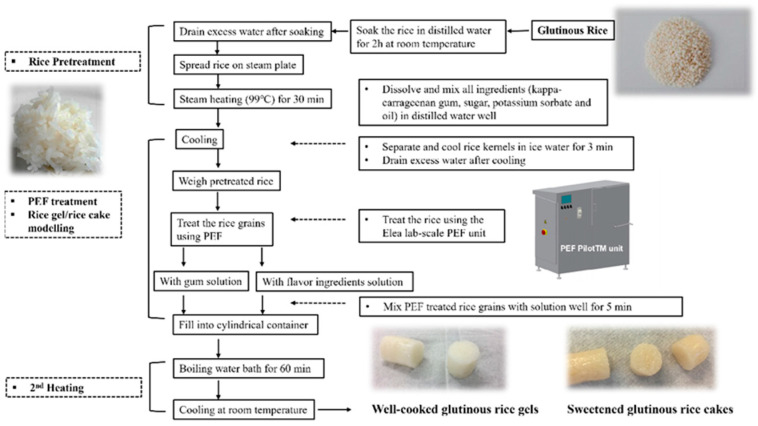
Schematic flowchart for the preparation of rice gel samples and rice cake samples made from glutinous rice grains with or without PEF treatment.

**Figure 2 foods-11-01306-f002:**
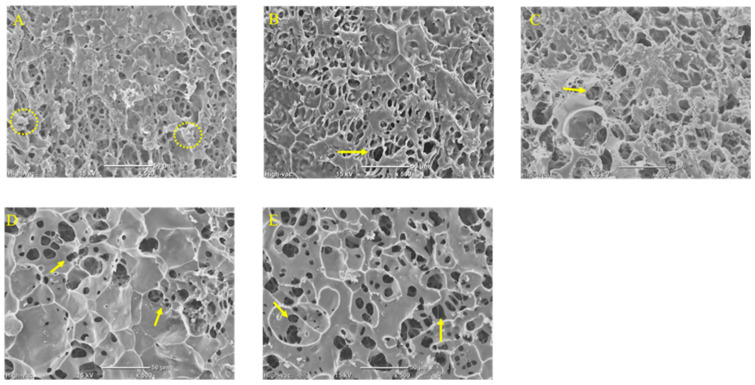
SEM images of steamed glutinous rice grain surface without PEF treatment (Control, (**A**)), and with 100 (**B**), 200 (**C**), 300 (**D**), and 400 (**E**) pulses of PEF treatment. Insufficient gelatinized starch particles were circled (**A**). Arrows indicate the pores or cavities on the PEF-treated rice grain surface. The SEM images captured for all samples provided a 500× magnification and used the scale bar of 50 μm.

**Figure 3 foods-11-01306-f003:**
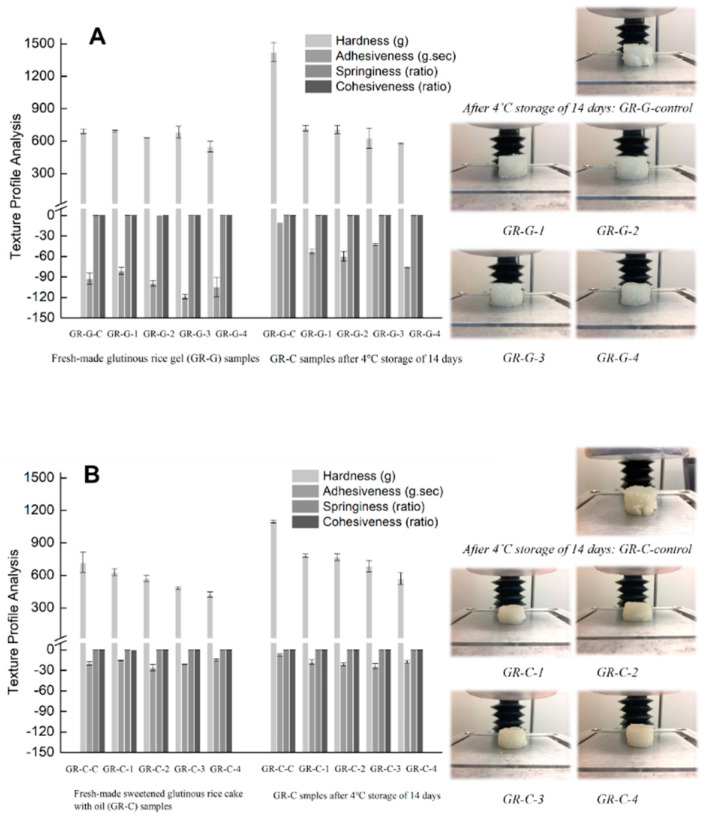
Textural properties of the freshly cooked and 14-day stored at 4 °C glutinous rice gel (GR-G) samples (**A**) and sweetened glutinous rice cake (GR-C) samples (**B**), and visual images of the stored GR-G and GR-C samples after TPA testing. The GR-G and GR-C samples were prepared using rice grains without PEF treatment (GR-G-control, GR-C-control), and with 100 (GR-G-1, GR-C-1), 200 (GR-G-2, GR-C-2), 300 (GR-G-3, GR-C-3), and 400 (GR-G-4, GR-C-4) pulses of PEF treatment. Each plotted point represented the average of five trials ± SD.

**Figure 4 foods-11-01306-f004:**
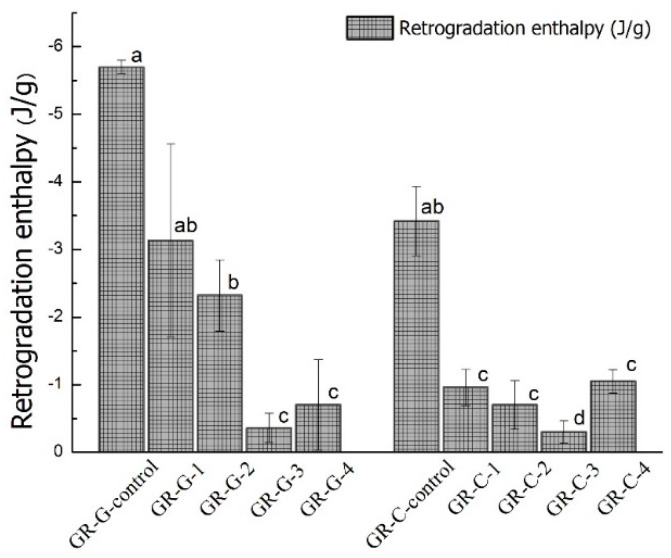
The retrogradation enthalpy after 14 days of storage at 4 °C of control glutinous rice gel sample (GR-G-control) and control sweetened glutinous rice cake (GR-C-control) and rice samples with 100 (GR-G-1, GR-C-1), 200 (GR-G-2, GR-C-2), 300 (GR-G-3, GR-C-3) and 400 (GR-G-4, GR-C-4) pulses of PEF treatment at 3 kV/cm. Mean values ± standard deviation of triplicates. Different letters on the bar graph indicated significant differences among the enthalpy of the samples (*p* < 0.05).

**Figure 5 foods-11-01306-f005:**
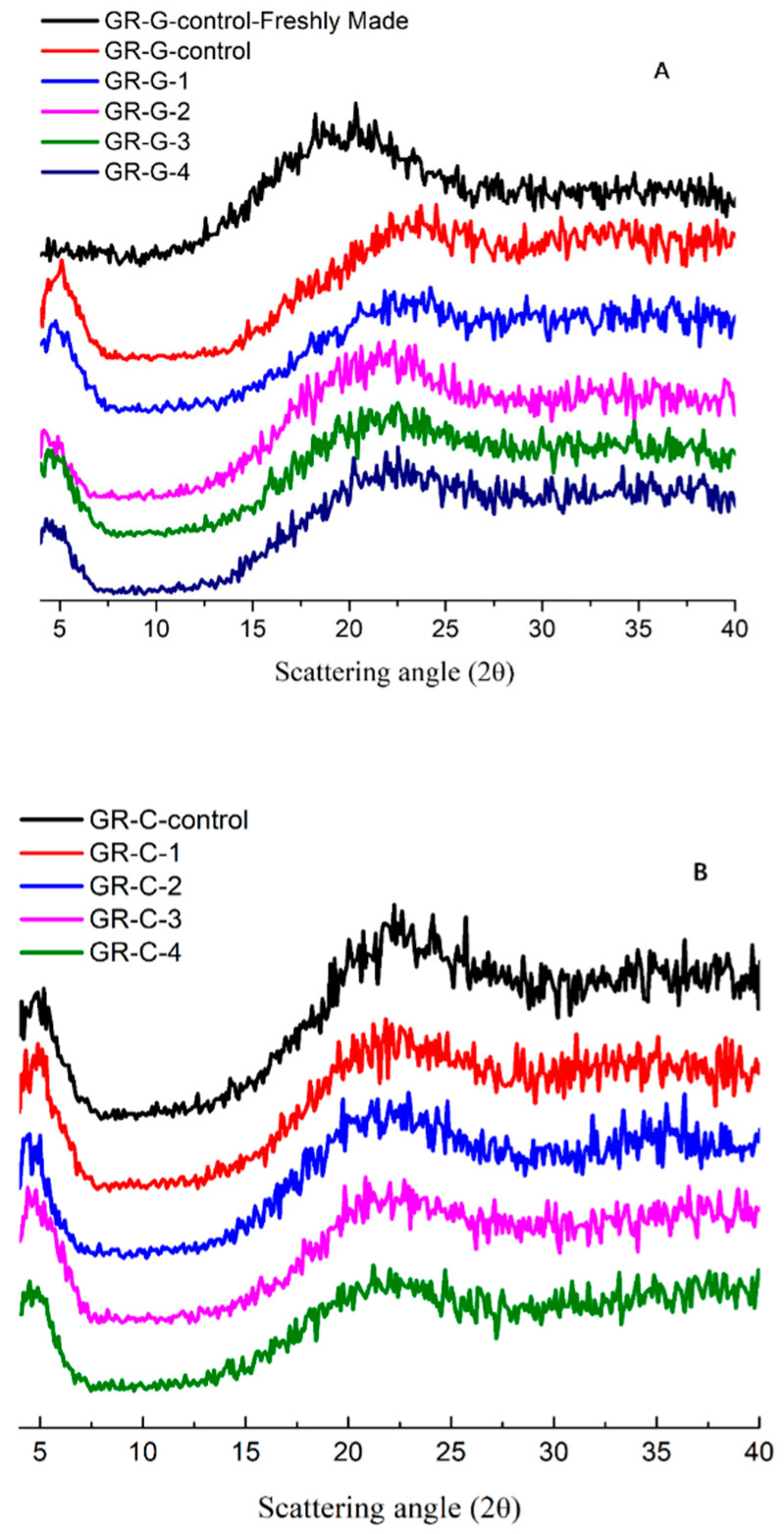
Separated X-ray diffractograms of glutinous rice gel sample (**A**) and sweetened glutinous rice cake samples (**B**). GR-G-control, GR-G-1, GR-G-2, GR-G-3, GR-G-4 represented that rice gel samples after 14 days of storage at 4 °C, made of 0, 100, 200, 300, 400 pulses of 3 kV/cm PEF treated rice grains, respectively; GR-C-control, GR-C-1, GR-C-2, GR-C-3 and GR-C-4 represented that rice cake samples after 14 days of storage at 4 °C, made of 0, 100, 200, 300, 400 pulses of 3 kV/cm PEF treated rice grains, respectively. GR-G-control-Freshly Made represented the freshly made control rice gel sample.

**Figure 6 foods-11-01306-f006:**
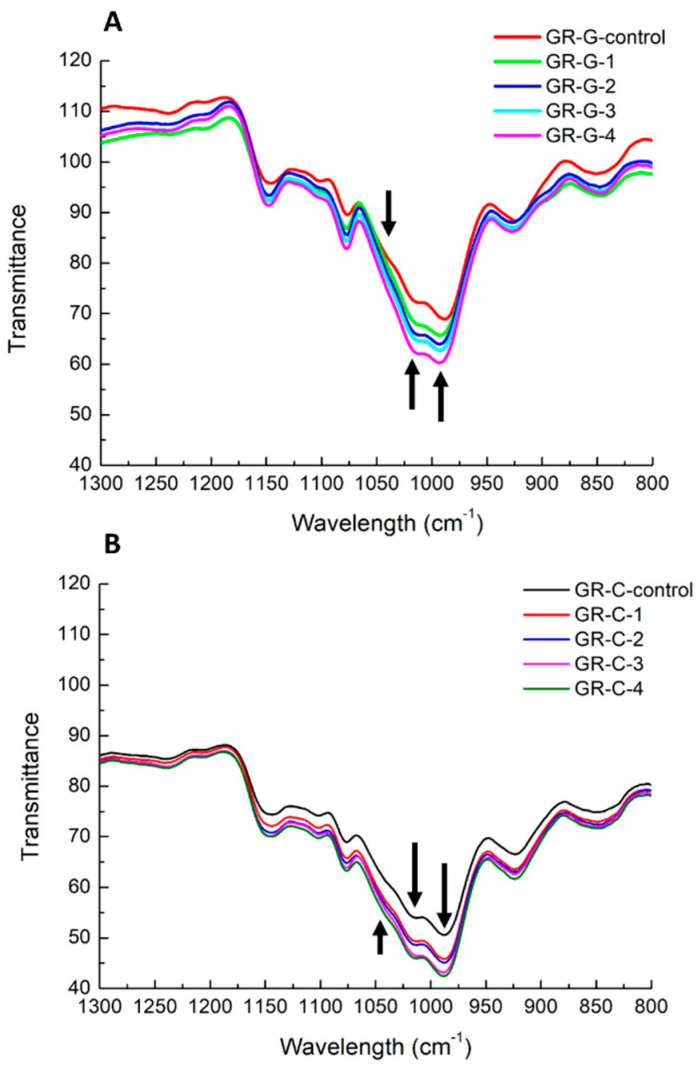
Overlay FTIR spectra of control glutinous rice gel (**A**) sample (GR-G-control) and control sweetened glutinous rice cake (**B**) sample (GR-C-control) and rice samples with 100 (GR-G-1, GR-C-1), 200 (GR-G-2, GR-C-2), 300 (GR-G-3, GR-C-3) and 400 (GR-G-4, GR-C-4) pulses of PEF treatment at 3 kV/cm after 14 days of storage at 4 °C at the regions of 1300–800 cm^−1^. Arrows indicate the peaks at the wavelength of 995, 1022 and 1047 cm^−1^ respectively.

**Table 1 foods-11-01306-t001:** Temperature and conductivity of rice grain suspensions before and after 100–400 pulses at 3 kV/cm of PEF treatment. Moisture content (*w*/*w*, wet basis) of fresh-made glutinous rice gel samples (GR-G) and sweetened glutinous rice cake samples (GR-C) with or without PEF treatment.

Temperature and Conductivity of PEF-Treated Rice Suspensions ^#^					
PEF Treatment					
Sample	Control	PEF-100 pulses	PEF-200 pulses	PEF-300 pulses	PEF-400 pulses
PEF treating pulses	0	100	200	300	400
*Rice grain/KCl suspensions before and after PEF treatment*					
Initial *T* (°C) ^#^	0.7 ± 0.0 ^a^	0.7 ± 0.1 ^a^	0.9 ± 0.0 ^a^	1.1 ± 0.2 ^b^	1.2 ± 0.1 ^b^
Final *T* (°C)	1.3 ± 0.0 ^a^	7 ± 0.7 ^b^	25.7 ± 0.3 ^c^	38.8 ± 0.7 ^d^	55.2 ± 0.4 ^e^
Initial θ (μS/cm)	250.0 ± 1.0 ^a^	250.0 ± 1.0 ^a^	260.0 ± 1.7 ^ab^	280.0 ± 0.7 ^b^	303.0 ± 3.0 ^c^
Final θ (μS/cm)	253.0 ± 0.3 ^a^	248.0 ± 3.1 ^a^	300.0 ± 3.0 ^b^	390.0 ± 1.0 ^c^	400.0 ± 2.5 ^c^
Final glutinous rice products made of PEF-treated rice grains *					
*Glutinous rice gel (GR-G)*					
Water content (%)	68.80 ± 0.01 ^a^	69.20 ± 0.01 ^a^	71.50 ± 0.01 ^ab^	72.20 ± 0.03 ^b^	76.30 ± 0.01 ^c^
*Sweetened glutinous rice cake (GR-C)*					
Water content (%)	63.10 ± 0.03 ^a^	63.50 ± 0.01 ^a^	64.80 ± 0.11 ^ab^	66.30 ± 0.01 ^b^	68.40 ± 0.01 ^c^

^#^*T*: temperature; θ: conductivity. *: Data are means of triplicate ± SD. Values followed by the same letters in the same raw do not differ significantly at the *p* < 0.05 level.

## Data Availability

The data presented in this study are available upon the request from the corresponding author. The data are not publicly available due to the confidential agreement with the sponsor.

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
