# Peer review of "Prevention of the Retrogradation of Glutinous Rice Gel and Sweetened Glutinous Rice Cake Utilizing Pulsed Electric Field during Refrigerated Storage"

_foods, 2022, doi:10.3390/foods11091306_

Round 1

Reviewer 1 Report

The paper is well written and covers a topic of high interest. Whilst numerous papers have looked into cell membrane electroporation the impact on biopolymers such as proteins or starch has been investigated to a lesser extent.

After a comprehensive literature review materials and methods are presented. The conditions of the PEF treatment should be specified in more detail to allow scaling of energy requirements to a potential industrial use, e.g. the energy input per kg of material should be calculated. The temperature levels before and after treatment are given, which are indicating energy levels applied but some heat losses may have occurred during the treatment time. A pulse repetition rate of 1 kHz is mentioned, which would result in very short treatment at 100 to 400 pulses. Please check.  

The results are well presented and discussed, conclusions drawn are justified.

Author Response

Please see attachment "Response to Reviewers Comment-Reviewer #1“

Reviewer 2 Report

The topic of your manuscript sounds interesting and an appropriate design. Overall information presented in this article provides a good foundation for future studies. 

Need to improve the abstract according to obtained results, if necessary then add some numerical results for better understanding.

Why author adopted the high number of electric pulses?

In case of differential scanning calorimetry (DSC) analysis, X-ray diffraction (XRD) pattern, and FTIR analysis authors only present the storage results. Why not authors present the results of fresh samples?

Author Response

Please see attachemnt "Response to Reviewers Comment-Reviewer #2"

Round 2

Reviewer 2 Report

no more comment